# Bone Metastases from Gastric Cancer Resembling Paget’s Disease: A Case Report

**DOI:** 10.3390/jcm11247306

**Published:** 2022-12-09

**Authors:** Hisaki Aiba, Tomoharu Nakazato, Hideo Matsuo, Hiroaki Kimura, Shiro Saito, Takao Sakai, Hideki Murakami, Jun Kawai, Shingo Kawasaki, Yasuhiro Imamura

**Affiliations:** 1Department of Orthopedic Surgery, Saishukan Hospital, 111, Shikatanishimuramae, Kitanagoya 481-0004, Japan; 2Department of Orthopedic Surgery, Nagoya City University, 1, Azakawasumi, Mizuho-cho, Mizuho-ku, Nagoya 467-8601, Japan; 3Department of Surgery, Saishukan Hospital, 111, Shikatanishimuramae, Kitanagoya 481-0004, Japan

**Keywords:** bone metastasis, gastric cancer, osteoblastic metastasis, Paget’s disease

## Abstract

Systemic osteosclerotic lesions are frequently caused by multiple bone metastases or systemic metabolic disorders. However, bone metastasis from gastric cancer is rare. Herein, we describe such a case, with radiographic and clinical findings resembling Paget’s disease. The patient was an 80-year-old Japanese woman with a history of early gastric cancer, treated by partial gastrectomy 2 years prior. The patient sought medical care for chronic low back pain. On imaging, systemic sclerotic lesions were observed throughout the spine and pelvis, with an increase in bone mineral density from 0.86 g/cm^3^ (2 years prior) to 1.38g/cm^3^ (current visit) in the lumbar spine. Elevated serum levels of osteoblastic and osteolytic markers were identified. A bone biopsy was used to confirm the diagnosis of metastatic gastric cancer. The patient was treated with TS-1 and denosumab, with normalization of abnormal metabolic markers and alleviation of the back pain. Bone metastasis is reported in only 10% of cases of gastric cancer and, thus, is relatively rare. Therefore, our case of gastric cancer recurrence presenting with mixed osteoblastic and osteolytic bone lesions similar to Paget’s disease is relevant to the report. Bone biopsy is necessary for an accurate diagnosis.

## 1. Introduction

Bone metastasis commonly occurs in advanced stages of cancer. Although bone metastases are common in prostate, breast, lung, and kidney cancer, these are a rare occurrence for gastric cancer [1]. Bone metastasis is generally associated with poor clinical outcome [2], with pain management and prevention of skeletal-related events (SREs) being the main treatment goals. Herein, we present a rare case of bone metastases from gastric cancer, presenting with radiographic features resembling those in Paget’s disease, which complicated the diagnosis.

## 2. Case Description

The patient was an 80-year-old Japanese woman who presented to our orthopedic clinic with complaints of chronic low back pain. The patient had a history of gastric cancer, a 10 × 8 mm tumor, type IIc, ulcerated lesion located on the posterior wall of the body of the stomach 2 years prior. The patient was treated with partial gastrectomy, with curative margins achieved and reconstruction performed using the Billroth-1 method. In the lesion, a diffuse sclerosing growth of poorly differentiated adenocarcinoma with signet ring cell was identified and diagnosed as signet ring cell carcinoma, pT2N0M0, pathological stage Ib. 

Radiographs obtained at the time of cancer diagnosis (2 years before) revealed normal bone mineral density (0.86 g/cm^3^, 79% of the young adult mean [YAM]) in the lumbar spine. On repeated investigation of the bone mineral density obtained 1 year later, the YAM of the spine had stabilized at 0.96 g/cm^3^ (YAM = 86%), with a sudden increase to 1.38 g/cm^3^ (YAM = 123%) at the time of the consultation for back pain. The patient was referred to our hospital to ascertain the etiology of these unexpected increases in spinal bone mineral density. At this time, sclerotic changes throughout the spine and pelvis were revealed (Figure 1a,b).

Physical examination revealed tenderness on palpation of thoracic, lumbar, and sacral vertebrae, with limited spinal motion due to the back pain. There was no evidence of paralysis or sensory loss in the extremities. Blood tests revealed elevated levels of alkaline phosphatase (ALP), tartrate-resistant acid phosphatase 5b (TRAP-5b), and procollagen-1 intact N-terminal propeptide (P1NP). Serum chemistry and blood counts for liver and kidney function were within normal range (Table 1). Electrophoretic distribution patterns of serum proteins and coagulation tests were also normal. 

Computed tomography (CT) images revealed the presence of mixed osteoblastic and osteolytic lesions in the clavicle, costal bones, vertebrae, and pelvis (Figure 2b,c), which had not been detected on CT images obtained 1 year prior (Figure 2a). There was no evidence of extra-skeletal metastatic lesions. Magnetic resonance imaging (MRI) revealed abnormal changes throughout the spine (Figure 3).

The rapid increase in bone mineral density combined with the abrupt increase in serum levels of biphasic bone metabolic markers was comparable to findings in Paget’s disease. Therefore, a bone biopsy of the sacral ala and a gastric mucosal biopsy via a gastrointestinal fiberscope were performed in the same patient to identify whether these findings were a result of Paget’s disease or further gastric cancer progression. Based on the signet-type deformities in the inter-trabecular areas, bone abnormality was confirmed as metastatic disease (Figure 4a). Recurrence of cancer in the residual postoperative stomach (group 5, poorly differentiated adenocarcinoma; Figure 4b,c) was also confirmed. Based on these findings, a final diagnosis of recurrent gastric cancer with multiple bone metastases was made. 

Treatment consisted of administration of antitumor compounds (tegafur + gimeracil + oteracil potassium, [TS-1], 50 mg, bis in die) and subcutaneous injection of denosumab (120 mg, monthly). Six months after treatment initiation, serum levels of bone turnover markers and tumor biomarkers had decreased (ALP, 52 U/L; P1NP, 35 ng/mL; TRAP-5b, 93 mU/dL; and CA19-9, 11 U/mL). Currently, 1.5 years after treatment initiation, the patient is leading a normal life, without progression of tumor symptoms. The abnormal radiographic changes in the lumbar spine and pelvis, including the abrupt changes in bone mineral density, persist.

## 3. Discussion

Bone metastasis from gastric cancer is uncommon [3]. Based on an autopsy study, the incidence rate of bone metastasis from gastric cancer is estimated at 13.4–15.9%, which differs from the incidence rate of 0.9–10% reported in clinical practice [4]. Bone metastasis from gastric cancers is predominant in the lumbar and thoracic vertebrae and costal bones [4]. Accordingly, back pain is a common symptom of metastatic gastric cancer [4]. In cases of metastatic gastric cancer, high serum levels of ALP and LDH were reported in 73.7% and 47.7% of cases, respectively [5]. Of note, elevated levels of tumor markers, including carcinoembryonic antigen (CEA) and CA19-9, are indicative of concomitant non-bone metastases of gastric cancer [4]. After detection of bone metastasis of gastric cancer, the estimated median survival time is 189 (range, 24–509) days [5]. Generally, the main objective of managing bone metastasis is controlling pain, maintaining physical activity, and preventing unexpected pathologic fracture or spinal cord compression with bone-modulating agents and/or radiotherapy [6].

Typically, metastatic bone lesions with osteoblastic changes from gastric cancer are rare, with fewer than 20 cases having been reported to date [7]. A review of gastric cancers by Okazaki et al. [8] identified that up to 80% of patients have poorly differentiated gastric cancers (signet ring cell carcinomas or poorly differentiated adenocarcinomas) [7]. The specific symptoms of osteoblastic metastasis of gastric cancers are hypocalcemia (in 28% of the cases) and elevated ALP levels. 

Disseminated carcinomatosis (DC) of bone marrow, defined as widespread bone metastases associated with hematological abnormalities (disseminated intravascular coagulation or microangiopathic hemolytic anemia), is an advanced stage of cancer [9]. Considering that the status of blood and coagulation was normal in our case, a diagnosis of DC was not applicable. However, if not treated appropriately, this case may have progressed to DC. 

Rapid proliferation of osteoclasts via expression of receptor activator of nuclear factor kappa-B ligand (RANKL) in gastric cancer cells plays an important role in the development of DC [10,11]. In addition, factors such as insulin-like growth factor (IGF), transforming growth factor-beta (TGF-β), bone morphogenetic protein (BMP), and platelet-derived growth factor (PDGF) play an important role in the development of cancer with the activation of osteoblastic lesions [7,9,12]. This process is different from the vicious cycle of osteolytic metastasis; it promotes the differentiation and function of osteoclasts and reduces osteoblast function [13]. Further analysis is necessary to elucidate the etiology of the biphasic activation of osteoblasts and osteoclasts.

In this case, because of the multiple, mixed types of bony changes, the differential diagnosis of abnormal alternation was osteoblastic metastasis (for example, breast cancer), fluorosis, renal dystrophy, or Paget’s disease. Paget’s disease, first described by James Paget [14], is characterized by excessive osteoclastic bone resorption and increased abnormal bone formation. The incidence rate of Paget’s disease differs by approximately 0.5–1% in Anglo-Saxon populations, compared to 1 per 2.8 million in Japan [15]. Histopathologically, there are abnormalities in osteoclast activity in Paget’s disease, such as increased number, size, and nuclearity (pyknosis) of osteoclasts and deep resorption lacunae (swallow-tail pattern), as well as excessive osteoblast activity, including formation of mosaics of woven bones, hyper-osteoblastosis, and isolated areas of poorly mineralized osteoids [16,17]. The levels of ALP and other bone turnover markers (TRAP-5b and P1NP) are considered important for screening and monitoring of bone metastatic disease. However, as we observed in this case, it is difficult to distinguish between Paget’s disease and metastatic tumors based on the levels of bone metabolic markers [18,19] and/or the imaging characteristics. Therefore, a bone biopsy is needed for differential diagnosis.

Denosumab, a monoclonal IgG antibody against RANKL, plays an important role in the management of tumors and prevention of SREs in patients with bone metastases from solid tumors via denosumab is recommended [20,21]. Denosumab can be used to treat several cancer-related complications, including hypercalcemia, mechanical pain due to bone fragility, and fractures [22]. In this case, denosumab was administered to suppress abnormalities in the expression levels of bone metabolic markers, which ultimately alleviated pain. Considering the poor prognosis of poorly differentiated gastric cancer, denosumab should be administered for its potential antitumor effects and to halt the vicious cycle of bone metastases.

## 4. Conclusions

We encountered a rare case of metastatic bone disease from gastric cancer presenting with mixed osteoblastic and osteolytic metastatic lesions resembling Paget’s disease. Bone biopsy was required for accurate diagnosis. 

## Figures and Tables

**Figure 1 jcm-11-07306-f001:**
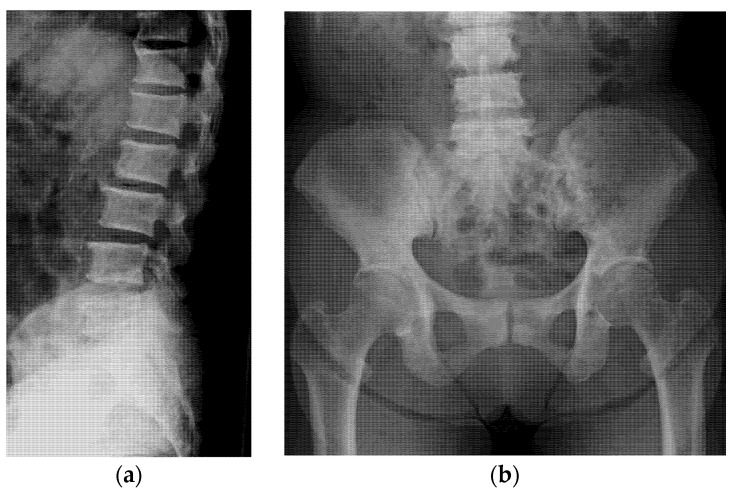
(**a**) Plain radiograph obtained before treatment showing sclerotic changes in the thoracic and the lumbar spine (lateral view). (**b**) The sclerotic changes extended to the pelvis but with no involvement of the femur (anteroposterior view).

**Figure 2 jcm-11-07306-f002:**
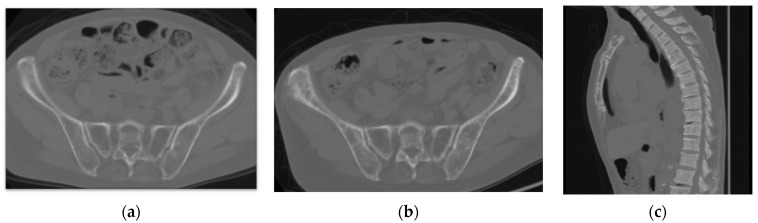
Computed tomography (CT) images. (**a**) CT image obtained 1 year prior to the consultation for chronic back pain, with no abnormal findings in the pelvis. (**b**) CT image of the same field of view obtained at the time of consultation for back pain revealed sclerotic changes in the sacral ala and iliac wing. (**c**) Sclerotic changes were also detected throughout the thoracic spine and sternum.

**Figure 3 jcm-11-07306-f003:**
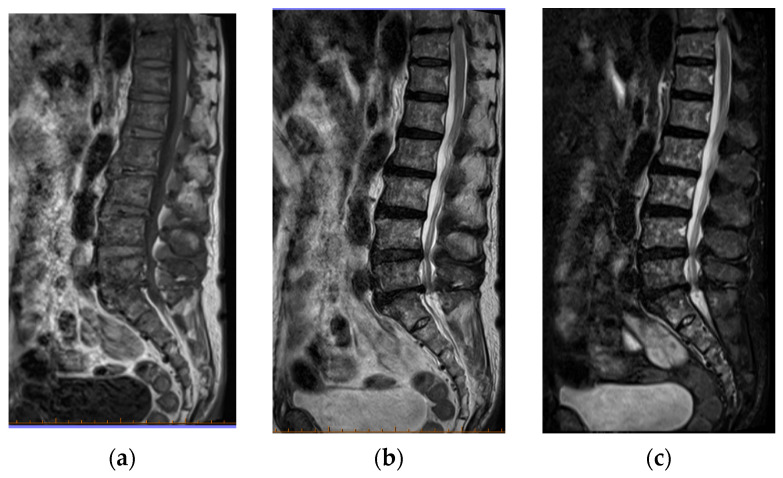
Magnetic resonance imaging (MRI) of the lumbar spine. (**a**) T1-weighted image. (**b**) T2-weighted image. (**c**) Fat-suppressed T2-weighted image. Abnormally low signal intensity was observed over the entire lumbar spine on T1- and T2-weighted images.

**Figure 4 jcm-11-07306-f004:**
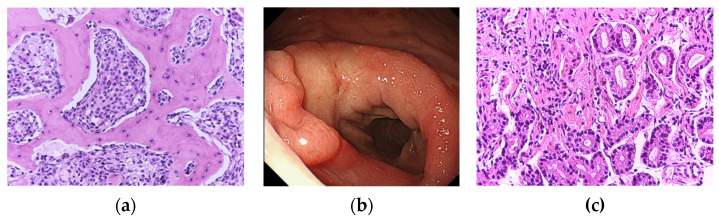
Pathological image of the bone lesion in the tissue collected by biopsy and endoscopic/ pathological images of the upper gastrointestinal region. (**a**) Proliferation of a signet ring, poorly differentiated tumor was detected in the inter-trabecular area (hematoxylin-eosin staining, ×100). (**b**) Abnormal reddish mucosa was observed around the anastomotic site. The biopsy specimens were collected from the slightly protuberant wall. (**c**) The pathological image from the biopsy. There was a similar proliferation of poorly differentiated tumor cells (hematoxylin-eosin staining, ×100).

**Table 1 jcm-11-07306-t001:** Results of laboratory tests conducted prior to treatment initiation.

Name of Items	Value	Normal Range	Units
AST	18	13–30	IU/L
ALT	9	7–23	IU/L
LDH	219	124–222	U/L
Creatinine	0.54	0.46–0.79	mg/dL
Na	144	138–145	mEq/L
K	4.8	3.6–4.8	mEq/L
Ca	9.7	8.8–10.1	mg/dL
Cl	108	101–108	mEq/L
WBC	4580	3300–8600	/mm^3^
Hb	10.9	11.6–14.8	g/dL
Platelet	18.4	15.8–34.8	×10^4^/mm^3^
CA19-9	44.1	0.0–37.0	U/mL
CEA	1.4	0.0–5.0	ng/mL
K	4.8	3.6–4.8	mEq/L
ALP	270	38–113	U/L
ALP1	0	0–0	%
ALP2	22	36–74	%
ALP3	78	25–59	%
ALP4	0	0–0	%
ALP5	0	0–0	%
P1NP	1200	26–98	ng/mL
TRAP-5b	3440	120–420	mU/dL
sIL-2	288	122–496	U/mL
PTH-intact	53	10–65	pg/mL

ALP, alkaline phosphatase; ALT, alanine transaminase; AST, aspartate aminotransferase; CA19-9, carbohydrate antigen; CEA, carcinoembryonic antigen; Hb, hemoglobin; LDH, lactate dehydrogenase; P1NP, procollagen-1 intact N-terminal pro-peptide; PTH, parathyroid hormone; sIL-2, soluble form of interleukin 2 receptor; TRAP-5b, tartrate-resistant acid phosphatase 5b; WBC, white blood cell.

## Data Availability

Not applicable.

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
