# Peer review of "Bone Metastases from Gastric Cancer Resembling Paget’s Disease: A Case Report"

_jcm, 2022, doi:10.3390/jcm11247306_

Round 1
Reviewer 1 Report
In the manuscript “jcm-1996775” the authors describe “an 80-year-old Japanese woman 15 with a history of early gastric cancer, treated by partial gastrectomy 2 years prior” who “sought medical care for chronic low back pain”. Imaging showed “systemic sclerotic lesions…… throughout the spine and pelvis, with an increase in bone mineral density……resembling Paget’s disease” and markers of bone turnover were increased. “Computed tomography (CT) images revealed the presence of mixed osteoblastic and osteolytic lesions in the clavicle, costal bones, vertebrae, and pelvis.” On biopsy “the diagnosis of metastatic gastric cancer” was confirmed. “The patient was treated with TS- 1 and denosumab, with normalization of abnormal metabolic markers and alleviation of the back pain.” As bone metastasis from gastric cancer is rare, the authors found “……relevant……” their case and conclude that histologic evaluation is “……necessary for an accurate diagnosis.”
Abstract
- For this reviewer the clinical (differential) diagnosis based on history, bone mineral density, imaging and laboratory data should be reported.
- Why the authors decide to use “young adult mean (YAM) in the lumbar spine” for bone mineral density?
Case description
- Please change “……ago” with ……before.
- If it is possible, report the histologic classification of the tumor according to the last WHO classification.
- Why the authors decide to use “young adult mean in the lumbar spine” and not age matched normal females for bone mineral density? Normal values should be reported.
- What the authors mean with “bone minerality”?
Discussion
- It can be shortened.
- Do not think the authors that development of DC sounds better than “formation of DC“?
- “However, further analysis is necessary to elucidate the etiology of the biphasic activation of osteoblasts and osteoclasts.”. This is true. However, please see for example doi: 10.3325/cmj.2021.62.270 and doi: 10.1038/s41413-020-00105-1.
- Please reformulate the sentence “metastasis was initially suspected to have originated from osteoblastic metastasis (e.g., breast cancer), fluorosis, renal dystrophy, or Paget’s disease.“ It is not clear for this reviewer.
- “However, as we observed in the case herein, it is difficult to distinguish between Paget’s disease and metastatic tumors on the basis of the levels of bone metabolic markers alone…”. ……also when they are combined with imaging (as this case report supports).
- “denosumab and anti-tumor agents were administered to suppress abnormalities in expression levels of bone metabolic markers to alleviate pain.” Is this sentence correct? Does it refer to denosumab alone?
Figure 2
- “comparable to osteopetrosis”… please see doi: 10.7860/JCDR/2015/13334.6348.
Figure 3
- Please change “intra-……” with inter-……
- Can you add a representative histological image of the recurrent gastric tumor?
- Signet-ring and not “signet-ring like”.
Author Response
Dear Reviewer1,
Thank you for your comments on my article. The altered parts were highlighted with yellow color.
Abstract
- For this reviewer the clinical (differential) diagnosis based on history, bone mineral density, imaging and laboratory data should be reported.
- Why the authors decide to use “young adult mean (YAM) in the lumbar spine” for bone mineral density?
In the Japanese guideline of the osteoporosis, we always YAM value for the diagnosis and the treatment effects. But, as your comment, it will be better to use the bone mineral density. I edited this point. (Abstraction and case description sections)
Case description
- Please change “……ago” with ……before.
I changed description.
- If it is possible, report the histologic classification of the tumor according to the last WHO classification.
I changed it with an opinion of a pathologist from a diffuse sclerosing growth of poorly differentiated adenocarcinoma and signet-ring carcinoma to stage pT2N0M0 to signet-ring cell carcinoma, pT2N0M0, pathological stage Ib.
- Why the authors decide to use “young adult mean in the lumbar spine” and not age matched normal females for bone mineral density? Normal values should be reported.
As abovementioned, I changed the description.
- What the authors mean with “bone minerality”?
I changed description.
Discussion
- It can be shortened.
- Do not think the authors that development of DC sounds better than “formation of DC“?
I changed description.
- “However, further analysis is necessary to elucidate the etiology of the biphasic activation of osteoblasts and osteoclasts.”. This is true. However, please see for example doi: 10.3325/cmj.2021.62.270 and doi: 10.1038/s41413-020-00105-1.
Thank you for your mention of good papers. As you said, sometimes, the mutual activation does not occur and contradictory results may occur depending on the types of the tumor and it’s microenvrionment.
- Please reformulate the sentence “metastasis was initially suspected to have originated from osteoblastic metastasis (e.g., breast cancer), fluorosis, renal dystrophy, or Paget’s disease.“ It is not clear for this reviewer.
We are sorry. The sentence was not correct. Then, the description was altered. “In the present case, owing to the multiple, mixed types of bony changes, the differential diagnosis of abnormal alternation was osteoblastic metastasis (e.g., breast cancer), fluorosis, renal dystrophy, or Paget’s disease.”
- “However, as we observed in the case herein, it is difficult to distinguish between Paget’s disease and metastatic tumors on the basis of the levels of bone metabolic markers alone…”. ……also when they are combined with imaging (as this case report supports).
I changed description to “However, as we observed in the case herein, it is difficult to distinguish between Paget’s disease and metastatic tumors on the basis of the levels of bone metabolic markers, and/or the imaging characteristics. Thus, bone biopsy needed for differential diagnosis.”.
- “denosumab and anti-tumor agents were administered to suppress abnormalities in expression levels of bone metabolic markers to alleviate pain.” Is this sentence correct? Does it refer to denosumab alone?
As you mentioned, denosumab alone is correct. I deleted it
Figure 2
- “comparable to osteopetrosis”… please see doi: 10.7860/JCDR/2015/13334.6348.
Thank you for the introduction of the osteopetrosis. As you said, the image is not similar to the osteopetrosis. I have just mentioned it as one of the differential diagnosis of bone sclerosing change. I deleted it.
Figure 3
- Please change “intra-……” with inter-……
- Can you add a representative histological image of the recurrent gastric tumor?
I inserted the image as figure 4c.
- Signet-ring and not “signet-ring like”.
I deleted it.
That's all. I greatly appreciated your review.
Reviewer 2 Report
I agree with the clinical conclusion that bone metastasis need to be biopsed.
I also remind the authors that Radiation therapy is a valid treatment for painful bone metastasis, regardless of the histology.
Author Response
Dear Reviewer 2,
Thank you for your comments on my article.
We understand the importance of radiation therapy.
We mentioned simply as the role of radiotherapy for the bone metastasis as following; Generally, the main objective of managing bone metastasis is controlling pain, maintaining physical activity, and preventing unexpected pathologic fracture or spinal cord compression with bone-modulating agents and/or radiotherapy.
Thank you for your consideration of our article.